# Impact of Polypropylene Fibers on the Mechanical and Durability Characteristics of Rubber Tire Fine Aggregate Concrete

**DOI:** 10.3390/ma15228043

**Published:** 2022-11-14

**Authors:** Arash Karimi Pour, Zahra Mohajeri, Ehsan Noroozinejad Farsangi

**Affiliations:** 1Department of Civil Engineering, University of Texas at El Paso (UTEP), El Paso, TX 79968, USA; 2Centre for Transportation Infrastructure Systems (CTIS), Department of Civil Engineering, University of Texas at El Paso (UTEP), El Paso, TX 79968, USA; 3Department of Civil Engineering, The University of British Columbia (UBC), Vancouver, BC V6T 1Z4, Canada

**Keywords:** mechanical characteristics, polypropylene fiber, rubber tire aggregates, ultrasonic pulse rate, sustainable concrete

## Abstract

In this research, the consequence of using rubber tire aggregates (RTA) on the durability and mechanical characteristics of polypropylene fibers (PF) reinforced concrete is evaluated. Fifteen concrete mixtures were produced and tested in the laboratory. RTA was utilized instead of fine natural aggregates (FNA) to the concrete at concentrations of 0%, 5%, 10%, 15%, and 20% by a volumetric fraction; also, the contents of PF in the concrete mixtures were 0%, 1%, and 2% by weight fraction. Finally, the following parameters were tested for all the mixtures: compressive and tensile resistances, fracture, changes in drying shrinkage, bulk electrical resistivity, elastic moduli, and resonance occurrences. The control sample was the one without RTA and PF. According to the results, by adding RTA to the mixtures, the shrinkage deformation amplified, but the PF addition caused a decrease in the shrinkage deformation. Furthermore, adding 0%, 5%, 10%, and 15% RTA, with 2% PF leads to an upsurge in the flexural resistance by 34%, 24%, 16%, and 6%, respectively, relative to the control sample without PF and RTA. Moreover, the fracture energy of mixtures increased by utilizing PF and RTA simultaneously.

## 1. Introduction

Compiling RTA around the world can cause a huge environmental risk if we cannot get rid of them properly. One efficient method, from both environmental and economic points of view, is to recycle these waste materials by using them as an aggregate for producing cement-based concrete [1,2,3,4]. Therefore, utilizing this waste as an agent for concrete production has become popular all over the world [5,6,7,8]. To see the effect of different contents of RTA, as a partial substitution for cement, FNA and coarse natural aggregates (CNA), on the characteristics of concrete mixtures, much research was directed. The influence of RTA on the mechanical performance of high-strength concrete was evaluated by Abdelmonem et al. [9]. RTA was used as a substitution for aggregates at different contents by volumetric fractions- 0%, 10%, 20%, and 30%. The high-strength concrete impact performance, slump, water absorption, density, fracture energy, compressive, and tensile resistances were measured. Based on the result of raising the RTA content, the adsorption of water decreased. Additionally, by adding more RTA, the flexural, tensile, and compressive performance of mixtures was decreased. As a result, adopting improvement methods to increase the strength of concrete can make it possible to use RTA in producing green concrete. Zhu et al. [10] conducted research on the comportment of high-performance RTA-incorporated concrete on the microstructural scale. Two methods were applied for that purpose, the first one was the Mercury intrusion porosimeter and the second one was scanning electron microscopy. The result illustrated that adding 5% RTA to the concrete mixtures leads to a 3.5% increase in their porosity. In addition, by increasing the RTA content, the pores’ size also increased, which leads to increasing the crack width which increases the need for solutions to improve the behavior of concrete.

In another study, Li et al. [11] assessed the characteristics of modified concrete with RTA on its surface. The findings revealed that the elastic moduli of the control mixture compared to the RTA-modified mixtures is lower. Consequently, the author applied a new correlation for electric modulus calculation for concrete mixtures according to RTA content. They recommended employing the additional materials for the RTA-incorporated concrete characteristics improvement. Based on the Strukar et al. research work [12], RTA as a partial substitution for FNA can be utilized, leading to deformation, ductility, and energy dissipation enhancement of the concrete mixtures. In another study, Li et al. [13] evaluated the concrete mechanical characteristics after adding RTA. According to the result, workability can be improved by adding RTA due to the particles’ hydrophobic nature and a proper mixture. So, by applying 0–10% RTA to concrete mixtures, the flexural resistance is decreased about 8–20%. Contrarily, RTA utilization had a disadvantageous impact on the initial density of concrete mixtures. Therefore, the incorporation of new materials is necessary to compensate for the negative influence of RTA incorporation. In addition, RTA concrete outperformed concrete made with natural particles in terms of freeze-thaw, sulfuric and sulfate attacks, and electrical, and abrasion resistances. Guo et al. [14] assessed the damage statuses of RTA concrete mixtures with a correlation of quantitative cloud images. To reach this goal, RTA was incorporated at various rates, ranging from 10% to 30%. Alternatively, one mix without RTA served as the control. An experimental test-based numerical model on the randomness of concrete with RTA was developed and validated. The consequence of RTA on the thermal stability and mechanical characteristic of concrete mixtures were examined by Záleská et al. [15]. A correlation between thermal conductivity and moisture content was found, and concrete sample properties, including secant and dynamic elastic modulus, and compressive and flexural resistances, were also examined. The weight and thermal conductivity of the concrete sample were decreased by using RTA. Additionally, the water transport capabilities of concrete mixtures with RTA were not significantly impacted. Moreover, temperatures up to 300 °C did not significantly alter the features of concrete related to the integration of RTA, but temperatures above 400 °C caused noticeable alterations. A recent thorough evaluation of the mechanical characteristics of concrete mixtures with RTA was published by Roychand et al. [16]. Therefore, it was investigated how various mechanical characteristics of RTA were impacted by the size of aggregate, treatment method, and replacement fraction. The findings showed that adding RTA to concrete has a reasonable impact on its hardened characteristics, but that the fresh characteristics considerably deteriorate.

According to Feng et al. [17], strain rates have an impact on how well RTA concrete flexes dynamically. The three-point flexural test was used to evaluate the concrete mixtures. Five RTA contents of 0%, 10%, 20%, 30%, and 40% were used to substitute FNA inside the concrete mixture. When the RTA percentage reached up to 30%, the concrete mixture exhibited a sharper strain rate compared to the control mixture; nevertheless, larger substitution contents led to no change in the strain rate. Additionally, compared to the regular concrete mixture, the sample contains 30% RTA as a partial replacement and has higher deformability and less rate of cracking. External confinement’s effect on the rubberized concrete’s compressive performance was considered by Chan et al. [18], as an improvement method to mitigate the negative influence of RTA. Using polymers that were fiber-reinforced significantly increased the axial performance of rubber-modified concrete, based on the results. Furthermore, constrained RTA concrete behaved very differently relative to the control concrete. The performance of a modified RTA mixture with fiber-reinforced polymers under load was therefore forecasted using a novel model. Wang et al. [19] recently evaluated the durability and mechanical behavior of RTA concrete modified with PF. The utilized RTA content was 10% and 15%. Moreover, to decrease the detrimental impact of RTA on the concrete mixtures, 5% PF were added to the mixes. The outcomes showed that employing fibers and RTA together might greatly increase fracture energy. Fibers were added to RTA concrete, which significantly improved its flexural resistance, drying shrinkage, and deformation.

The subject of another research by Farhad and Ronny [20] was the investigation of the RTA concrete characteristic containing PF. RTA with sizes 2–5 mm were utilized as a substitution for 20% FNA. The content of fibers is determined considering the type of fibers and volumetric ratios of 0.1–0.25%. The fresh characteristics of the mixtures were examined by the workability and the J-ring tests, and the hardened characteristics were evaluated by the compressive stress–strain, compressive, and tensile resistance tests. According to the findings, by increasing the content of fibers, the detrimental impact on the rheological characteristics of the concrete that are modified by fibers increases. Regarding mechanical characteristics, as the content of PF was raised, the compressive resistance lessened. In another study, the mechanical characteristics of cement-based RTA mortars that were modified by fibers were evaluated by Nguyen et al. [21]. To prevent cracking at very early stages, a special type of steel fiber with high bond properties with 20, and 30% volume content was applied. To achieve tensile strength, strain capacity, and behavior after residual peak, direct tensile experiments were carried out. Moreover, for obtaining Young’s modulus and compressive resistance, compressive experiments were conducted. According to the results, although adding RTA to concrete mixtures causes a reduction in compressive and tensile resistance and elastic modulus, it leads to an increase in strain capacity.

## 2. Research Significance

Literature reviews illustrate that a significant number of research studies have been performed on the evaluation of RTA-incorporated concrete. The application of RTA-based concrete has been limited to non-structural functions because of some of its significant weaknesses, such as early cracking, low strength, and stiffness, due to the inadequate bonding between RTA and concrete paste. Additionally, the previous finding showed that the incorporation of RTA to produce eco-friendly concrete led to reducing the mechanical characteristics of concrete. Therefore, using other improvement techniques is necessary to improve the performance of green concrete having RTA. In this regard, previous investigations showed that the incorporation of fibers significantly enhanced the mechanical characteristics of various types of concrete [22,23,24,25,26,27,28,29,30]. For this aim, PF was utilized to enhance the behaviors of concrete made with RTA. One noticeable point is that in most of the earlier investigations, the volume content of PF in the concrete mixtures was less than 1.5%, with high contents of RTA. Moreover, the RTA and PF effects on the mechanical experiments, freezing thaw, compressive and tensile strength, and slum were investigated. Furthermore, the modified concrete behavior after PF addition was assessed. Consequently, the concrete’s mechanical and long-lasting behavior that is affected by PF and RTA addition was investigated. To achieve this goal, various PF and RTA were used to measure their effects on mechanical experiments, such as drying shrinkage, ultrasonic pulse velocity, fracture energy, bulk electrical performance, freeze-thaw damage, and length changing.

## 3. Materials and Methods

### 3.1. Polypropylene Fibers

To produce fiber-reinforced mixtures, 12 mm length PF were introduced at three volumetric fractions of 0%, 1% and 2%. The tensile resistance, density and melting point of used fibers are 360 MPa, 890 kg/m^3^ and 160 °C, correspondingly.

### 3.2. Rubber Tire Aggregate

RTA comes from waste tires with particle sizes of 5–30 mm. RTA is used as a fractional substitution for FNA at five volumetric fractions- 0%, 5%, 10%, 15%, and 20%. Based on the Si et al. [31] study, RTA was submerged for 30 min in a NaOH solution with 40 g/L density and after that was washed and air-dried. Figure 1 shows a sample of utilized aggregates.

### 3.3. Concrete

Ordinary Portland cement was utilized for concrete production in a drum-type mixer based on the ASTM C150/C150M [32]. At first, for one minute, all dry particles were mixed, after that PF was added, and mixed for some minutes. Then, for another minute, mixing continued with two-thirds of the water addition. Finally, the rest of the water and superplasticizer were blended. Table 1 illustrates the cement’s chemical and physical characteristics. Moreover, the aggregates’ mechanical characteristics are provided in Table 2, as per ASTM D2419 [33]. Figure 2 presents the aggregates’ gradation graph. A total of five volumetric contents of RTA were utilized 0%, 5%, 10%, 15% and 20%; also, three weight contents of PF were incorporated: 0%, 1% and 2%. Accordingly, the mixture design can be seen in Table 3.

## 4. Results and Discussion

### 4.1. Slump

Slum flow is applied to evaluate the fresh concrete behavior. The goal of this is to test the sample’s workability determination based on the ASTM C143 [34]. All the concrete mixture outcomes are shown in Figure 3. From the workability point of view, the size of all slums for fresh mixtures is in the range of 9.8, and 19.7 cm. Moreover, raising the RTA content, caused a decrease in the concrete slump, specifically at 20% rubber content and 2% PF, where the slump dropped roughly 49%. This phenomenon is due to the higher friction between the RTA and other concrete components. The noticeable point is that adding PF to mixtures did not affect the concrete air void significantly. The RTA naturally repel water, and as a result, this leads to an increase in the amount of air trapped in the mixture. Additionally, when friction increases at the contact surface of RTA, PF leads to PF gathering and air trapping. Alternatively, in the absence of PF, the reduction in the slump was lower. As a result, by adding 20% RTA, the slump reduction was about 18% in the absence of PF. Furthermore, adding 20% RTA with 1% PF together leads to a 31% decrease in a slump.

### 4.2. Compressive and Tensile Resistances

After 28 days, tests were performed using the ASTM C496 [35] to determine the compressive and tensile resistances. The hydraulic jack maximum load was noted to calculate the tensile resistance, as illustrated in Figure 4. Three cylindrical samples with dimensions 300 mm by 150 mm were utilized for this experiment, and the load was applied to specimens. For this test, the load was applied at 1 mm/s. The average of the three specimens’ outcomes was considered as these characteristics value that is calculated by Equation (1).
(1)$$f_{t}=2P/\left(\pi DL\right).\;$$
where $f_{t}$, *P*, *D* and *L* are the tensile resistance, the failure point, the samples’ diameter, and the cylinder height, respectively.

The tensile resistance of RTA-based concretes that were modified with PF is illustrated in Figure 5 and Figure 6. It should be noted that in Figure 6, the increase in the area enclosed by the graphs generally indicates the level of effectiveness. The tensile resistance is determined by the adhesion in the contact area of cement components and particles, and also mixture strength. PF enhances tensile resistance noticeably by reducing the maximum crack width. Consequently, by applying just 20% RTA, the value of tensile strength dropped 23%, whereas, utilizing 2% PF in the concrete mixture improved the tensile strength by about 38%, also, prohibited the tensile resistance decrease when RTA is employed as a fractional substitution for FNA. Furthermore, by applying 2% PF with the RTA contents 0%, 5%, 10% and 15%, the rubberized concrete enhancement was about 28%, 18%, 13% and 5%, respectively. The tensile resistance is significantly affected by utilizing PF because the fragile cement paste cannot bear tensile tension, and in that case, the cement matrix quality determines the crack’s beginning. Therefore, the main cement paste crack typically appeared at the peak load. However, once the crack had already begun, the quick fracture spreading was significantly controlled by the bridging action of fibers. As a result, PF presence successfully prevented crack growth by enhancing the splitting tensile strength. Moreover, Figure 7 illustrates the PF’s favorable effect on mixtures with various rubber content when it came to tensile resistance. Under axial tensile load, and when PF was applied, cracking just took place on the concrete surface, and the whole sample did not collapse. However, the samples were kept from completely disintegrating using RTA as NFA. Additionally, specimens lacking PF split into two pieces at failure; however, by adding PF, this behavior was averted. RTA usage reduces concrete’s tensile resistance, although this can be made up for by PF addition. The use of fibers lengthens the concrete’s failure time by making a bridge between any potential fractures that enhances the tensile resistance. As a result, as per ASTM C496 [35] to evaluate the specimens’ function in the hardened state, the outcomes of measuring how PF and RTA affected the specimens’ compressive resistance are depicted in Figure 8. Notably, the mixtures’ age upon curing is also considered. Three cylindrical shape samples measuring 150 mm by 300 mm were subjected to a hydraulic jack for this goal.

As shown in Figure 8, employing RTA causes concrete’s compressive resistance to decrease; nevertheless, this characteristic was significantly enhanced when PF was used. As compared to natural aggregate, RTA has a larger Poisson ratio and lower stiffness, which contributes significantly to the strength decrease. The deterioration of strength may also be caused by poor adherence to cement paste and RTA. The addition of RTA also had an impact on the stability of the concrete samples, causing the elastic modulus to drastically decrease and, therefore, the entire concrete mixture stiffness decrease. Consequently, adding 2% PF with 0%, 5%, 10%, 15% and 20% RTA improved the concrete’s compressive resistance in 7 days by nearly 24%, 29%, 24%, 25% and 22%, correspondingly. As a result, comparable trends were seen at 14 and 28 days. In addition, as the curing time grew longer, the PF impact on the RTA-based concrete’s compressive resistance increased. A similar decreasing pattern can be observed in tensile resistance reported by Wang et al. [36] for concrete having only RTA. In addition, by utilizing high content of RTA, Rubber aggregates’ decrease impact will decline.

New relationships that considered the effects of both RTA and PF were developed in accordance with the findings of the compressive and tensile resistance tests, as shown in Figure 9. To forecast the tensile and compressive behaviors of RTA-based concrete, it is possible to utilize the suggested formulation with good values for fitting and R^2^ greater than 92%, as provided in Equation (2). In this equation, $f_{t}$. and $f_{c}$. indicates the splitting tensile and compressive strengths, respectively. Regarding Figure 9, as anticipated, the compressive and tensile resistances both increased concurrently.
*f_t_ =* 0.25*f_c_*^0.7^(2)

### 4.3. Flexural Resistance and Fracture Energy

According to the JCI-002-2003 standard, the fracture energy of the sample was calculated [37]. Therefore, after 28 days, single-edge notched beams with the dimensions 102 × 102 × 381 mm were fabricated and evaluated using a three-point bending arrangement (Figure 10). The notch’s width and depth are equal to 5 mm, and 30 mm, correspondingly. Loading continued until the samples’ complete failure. Equation (3) was used to calculate the specimens’ flexural resistance, where *L, F, h,* and *b* stand for the span width, maximum load, broken ligament height, and broken ligament width (area above the notch), respectively. The findings are shown in Figure 11.
(3)$$\sigma =3FL/\left(2bh^{2}\right).\;$$

Therefore, the JCI-002-2003 standard’s proposed formula [37] was used to calculate the fracture energy using Equation (4), where, where $G_{F}$, $W_{0}$, $w_{1}$, $A_{i}$, $m_{1}$,
$m_{2}$, *g*
and $CMOD_{c}$. indicates the fracture energy, the area below the load-displacement curve up to the failure of the specimen, work performed by deadweight of the specimen and loading, the area of the broken ligament, the mass of the samples, load point width, the whole length of the sample, the mass of the jig not devoted to the loading apparatus but located on the sample until failure, gravity and load-deformation at the time of rupture, correspondingly. The fracture energy is demonstrated in Figure 12.
(4)$$\left\{\begin{array}{c} G_{F}=\frac{0.75W_{0}+w_{1}}{A_{jig}}\\ w_{1}=0.75\left(\frac{S}{L}m_{1}+2m_{2}\right)g\times CMOD_{c} \end{array}\right.$$

As shown in Figure 11, introducing PF somewhat made up for the decrease in flexural resistance that occurred when RTA was substituted for the samples. As a result of PF’s bridging function, increasing particle connection, and improvement of concrete’s tensile and compressive resistances, particularly when RTA was not employed, PF significantly increased the flexural resistance of the specimens. The inelastic fracture load of rubber-based concretes in the absence of PF quickly decreased; however, once PF was added, everything changed completely, and the flexural resistance significantly increased. Therefore, by adding just 2% PF the flexural resistance, was improved by about 34%.

Furthermore, the addition of 2% PF together with 0%, 5%, 10%, and 15% RTA caused an improvement in the flexural resistance of the concrete mixtures by 34%, 24%, 16%, and 6%, correspondingly. In contrast, when RTA and PF were utilized in the mixture together, rubber particles increased the mixture’s fracture energy (Figure 12). Following the test, the failure patterns of the various specimen kinds were also examined. Only a little single fracture was observed in the control sample, which lacked RTA particles and PF. However, when PF and RTA were combined, the failure mechanism and the crack formation scenario were clearly altered. Additionally, the considerable opening of the crack mouth in modified mixtures with PF demonstrates the fiber bridging effect; yet, the sharpness of the fracture did not alter in comparison to the control sample. RTA and PF were combined, and the stress-relieving properties of elastic rubbers changed the crack’s formation course (Figure 12). As a result, ordinary concrete crumbled once the crack appeared and swiftly spread due to the low tensile resistance of the paste, demonstrating small fracture energy. Though, afterwards, the fracture started, the pull-out procedures of the arbitrarily positioned PF used a sizable amount of energy, greatly enhancing the fracture energy of the control mixture. The fracture energy was also raised with the inclusion of RTA, even while the flexural resistance was decreased with the addition of RTA and the reduction was elevated by increasing the RTA percentage in RTA-based concrete mixtures that were modified by PF. Figure 13 shows the flexural failure of specimens having RTA and PF.

The relations among flexural resistance and fracture energy, as well as among flexural and compressive resistances, are shown in Figure 14 and Figure 15. As shown in Figure 14, employing PF reduced the slope of the figures. The very precise models given to estimate the link between fracture energy and flexural resistance with various RTA concentrations are shown in this image. In addition, Figure 15 projected formulas among specimens’ flexural and compressive resistances have strong R-square values, indicating an upright fit, as provided in Equation (5). In this equation, $f_{r}$. and $f_{c}\;$denote the flexural and compressive strengths of concrete samples, respectively. Additionally, as anticipated, the flexural resistance increased at the same time as the compressive resistance.
*f_r_ =* 0.31*f_c_*^0.84^(5)

### 4.4. Ultrasonic Pulse Velocity

As per the Guo et al. study [38], the ultrasonic pulse velocity measurement was conducted after 28 days. A sample with a thickness of 30 mm and a diameter of 100 mm was made for this test. The top and bottom outsides of samples were applied to two Olympus 5070 transducers at a frequency of 0.5 MHz, and the average of three samples was used to calculate the pulse speed. The findings are shown in Figure 16, which demonstrates that employing both PF and RTA has a significant negative impact on this attribute. Additionally, the value of the ultrasonic pulse velocity decreases in the underdeveloped area when 20% RTA is applied. So, the ultrasonic pulse velocity is decreased by roughly 21%, 28%, and 31%, respectively, when 20% RTA and 0%, 1%, and 2% PF are used. Since the RTA’s elastic moduli are significantly lower than that of FNA and the cured cement matrix. Conversely, RTA increased the number of air spaces in the solid slides, which may have an impact on the pulse rate. Employing RTA causes concrete’s ultrasonic pulse velocity to decrease; nevertheless, this characteristic was significantly enhanced when PF was used. As compared to natural aggregate, RTA has a larger Poisson ratio and lower stiffness, which contributes significantly to the ultrasonic pulse velocity decrease. The deterioration of strength may also be caused by poor adherence to cement paste and RTA. The addition of RTA also had an impact on the stability of the concrete samples, causing the ultrasonic pulse velocity to decrease. As a result, adding RTA to ordinary cement mix reduced its ultrasonic pulse velocity. Previous research has not employed this test to assess the behavior of PF-reinforced RTA, demonstrating its originality.

### 4.5. Drying Shrinkage

As per ASTM C157 [39], concrete’s drying shrinkage performance was evaluated. For this test, 25 mm × 25 mm × 286 mm samples were tested at 1, 3, 5, 7, 11, 14, and 21 days of curing. After 24 h of curing, the samples were taken out of the molds and kept in the curing chamber at a temperature of 23 °C and relative humidity of 50%. Each sample’s starting length was measured right after demolding, and Figure 17 shows how the length varies over time. The samples’ lengths changed as the amount of RTA grew, as seen in Figure 17. Due to the lessened stiffness of RTA when compared with NFA, RTA could be easily deformed under the internal drying shrinkage stress; however, incorporating PF decreased the shrinkage length deviations and the amplified drying shrinkage caused by RTA could be properly limited with the application of PF. Furthermore, the length variations got worse at 14 days of cure and got worse till 21 days. As a result, utilizing RTA of 5%, 10%, 15%, and 20% increased the length variation at 28 days by 10%, 14%, 21%, and 27%, correspondingly. However, when 2% PF was added along with 0%, 5%, 10%, 15%, and 20% RTA, shrinkage decreased by roughly 37%, 28%, 21%, 20%, and 15% in comparison to the control specimen without PF and RTA. In addition, utilizing 2% PF has a greater impact than using 1% PF on minimizing the amount of shrinkage length changes.

### 4.6. Bulk Electrical Resistivity

Hamed et al. [40] developed a novel approach for determining concrete’s bulk electrical resistivity in 2015. For this aim, cylindrical samples with a diameter of 102 mm and a height of 204 mm were subjected to a frequency between 10 Hz and 10,000 Hz. The findings of three samples were averaged, and the results are shown in Figure 18. A total of 10% of RTA is the ideal amount to use to produce the highest possible bulk resistivity value. Therefore, by adding 2% PF and 10% RTA, this characteristic was enhanced by around 12%. The bulk resistivity is significantly improved by using PF, particularly when more than 10% RTA is used in place of natural aggregates. Furthermore, when more than 10% RTA was added, the resistance significantly decreased. The pathways of the pore solution can be obstructed by the rubber aggregates, Increasing the bulk electrical resistivity; nevertheless, employing extra RTA than 10% increased the absorbency within the mixture and increased slump. As a result, utilizing RTA of 5%, and 10% increased the bulk electrical resistivity of concrete by 8% and 10%, correspondingly. However, when 2% PF was added along with 0%, 5% and 10% RTA, the bulk electrical resistivity of concrete amplified by roughly 7%, 9.5% and 16%, in comparison to the control specimen without PF and RTA.

### 4.7. Expansion

The quick alkali-silica response test to gauge concrete’s length expansion is described in ASTM C1260 [41]. The specimens utilized in this test are the same size as those that were employed in the shrinkage test. To cure the samples, tap water was used in a container that was positioned in an oven set at 80 °C for 24 h before the samples were demolded. Following the early measurement, the samples were submerged in a second container containing a 40 g/L NaOH solution. Figure 19 shows the test marks as a function of the amount of PF and RTA used. After 7 days, the expansion rate significantly increased. Furthermore, the length expansion changes were significantly reduced because of the use of RTA. In addition, after 11 days, the length expansion changes were significantly slowed by 20% RTA (Figure 19e). Because of its bridging action, applying 2% PF delays the length expansion of mixtures having PF. The fibers may constrain the cement matrix since it tended to expand, which would prevent further growth of the expansion. Additionally, it should be highlighted that by boosting the amount of RTA, the impact of PF on the alkali-silica response growth is reduced. Consequently, when 20% RTA was employed, there was no discernible difference in length expansion between reinforcing concrete with 2% PF and 1% PF.

### 4.8. Freeze-Thaw Resistance

As per ASTM C666-15 [42], concrete specimens measuring 76 mm by 102 mm by 381 mm were created and kept in a freeze-thaw chamber. Therefore, after 28 days of underwater cure at 220 °C, the samples’ early mass weight, length, and resonance frequency were noted. During the test, length variations and the dynamic elastic modulus were assessed under different freeze-thaw cycles (0, 36, 50, 100, 150, 200, 250, and 300). As a result, Figure 20 shows the relative modulus of dynamic elasticity. The increase in dynamic elastic modulus was a sign that the concrete specimens generated for this evaluation were of good quality and freeze-thaw resistant. As shown in Figure 20, utilizing RTA with up to 10% raises the relative dynamic elastic modulus, but using RTA with more than 10% causes a large decrease in this parameter. When compared to the original state, the hydration created a more solid structure inner of the specimens which results in improving the relative dynamic elastic modulus. For 10RTA-0PF specimens, the relative dynamic elastic modulus significantly increased after 50 cycles. Alternatively, when 15% or 20% RTA were utilized, adding PF decreased the dynamic elastic modulus (Figure 20c). It should be mentioned that the dynamic modulus changes significantly up to 35 cycles. The dynamic elastic moduli remained constant after 45 cycles, and a similar trend was seen for samples with various PF and RTA amounts. Therefore, using 10% RTA with 0%, 1%, and 2% PF, respectively, raised the property’s maximum value by around 48%, 56%, and 51%. Almost the same results were reported by Alsaif et al. [43]. When RTA aggregates were used, the dynamic elastic modulus reduces by 10% in comparison with the control sample. 

Moreover, the specimens’ resonance frequency was evaluated. The resonant frequency of the samples during the freeze-thaw testing is demonstrated in Figure 21. The resonant frequency was dramatically lowered using RTA and PF. Furthermore, there were significant fluctuations in resonance frequency up to 50 cycles. However, when 15% and 20% RTA were used, this property did not stabilize until after 100 and 150 cycles, respectively. A concrete’s resistance to freeze-thaw cycles is shown by an increase in the dynamic modulus of elasticity and a decrease in resonance frequency. Because RTA is considerably less rigid than traditional fine aggregate, the resonance frequency of concrete samples has been reduced. Additionally, the RTA may allow more air to enter the concrete samples, increasing the amount of air space in the cured concrete specimens.

### 4.9. Freeze-Thaw Damage Durability Aspect

As per the consequences of the dynamic elastic moduli utilizing the next equation, durability was taken into consideration:(6)DF=PN/M

The durability feature of the samples, the relation dynamic elastic moduli at N cycles, the number of cycles until P ranges the detailed lowest rate for stopping the examination, and the quantified number of cycles until the experience is to be stopped is represented by the letters *DF, P, N*, and *M*, correspondingly. The durability of the samples is shown in Figure 22. As the freeze-thaw performance is improved, the durability factor increases. In relation to durability, 10% represents the RTA’s ideal value because beyond that point, it significantly decreased. The inclusion of RTA has enhanced the air content, which might enhance freeze-thaw resistance. Additionally, as the bulk electrical resistivity result indicates that the transport property of the 10RTA-0PF sample is the lowest, samples containing 10% RTA may associate with improved durability. Additionally, incorporating PF increased durability, particularly when 2% PF was applied. Additionally, once 2% PF was added with 0%, 5%, and 10% RTA to create concrete, the durability factor increased by 13%, 31%, and 37%.

### 4.10. Length Change

At each exposure stage, the length variations of several samples were documented, as can be shown in Figure 23, where the length underwent a significant shift for up to 50 cycles. Due to continuous cement hydration, autogenous shrinkage occurred whereas samples were kept in the freeze-thaw chamber. The specimens’ length gradually increased for the number of cycles less than 50, and then decreased by increasing the number of the freeze-thaw cycle. Additionally, by boosting the PF content, the specimens’ length reduction became more pronounced. Additionally, by adding more RTA, the length of the samples was further reduced. Furthermore, by boosting the content of RTA, the variations in specimen length recovered less. Specimens made with RTA exhibit more significant shrinkage changes than those made with natural aggregates. The specimens’ length roughly stabilized after 150 cycles.

## 5. Conclusions

The current investigation evaluated the characteristics of concrete having PF and RTA. Fifteen mixes were created with this objective in mind. Five ratios of RTA (0, 5, 10, 15, and 20%) were employed instead of FNA. The concrete mixes also included PF at three different volume contents: 0%, 1%, and 2%. Then, measurements were made of the specimens’ workability, compressive and tensile and flexural resistances, ultrasonic pulse velocity, shrinkage, electrical resistance, dynamic elastic moduli, and resonance frequency. The subsequent conclusions could be made based on the achieved findings:Workability was reduced by adding more RTA, especially when a combination of 20% RA and 2% PF was utilized, which resulted in a slump reduction of nearly 49%. Conversely, when PF was not utilized in the concrete mixtures, workability was reduced less. So, this property was reduced by about 18% when only 20% RTA was used. Additionally, utilizing 1% PF and 20% rubber reduced slump by about 31%.Utilizing RTA reduced the concrete’s tensile resistance. While PF greatly enhanced this feature. Due to the minor value of RTA relative to FNA, the tensile strength decreased by 28% when 20% RTA was used. Though, incorporating 2% PF amplified the tensile resistance by 5% and prohibited the drop of tensile resistance because of the fiber-bridging character in preventing the quick cracks growth.Compressive resistance of concrete was decreased when RTA was used; however, this characteristic was significantly increased when PF was incorporated. In comparison to concrete produced with merely 0%, 5%, 10%, 15%, and 20% RTA combining 2% PF increased the concrete’s 7-day compressive resistance by about 24%, 29%, 24%, 25% and 22%, correspondingly.By employing RTA, the specimens’ flexural resistance was decreased while PF significantly enhances the flexural performance of the specimens, particularly when RTA was not utilized, due to its bridging function, which increases particle connection and elevates concrete’s tensile and compressive resistances.Incorporating SF and RTA significantly declined the ultrasonic pulse velocity due to the elastic modulus of RTA being lesser than that of the concrete paste and FNA, when 20% RTA was employed, the ultrasonic pulse velocity decreased in the weak zone.Due to the RTA’s lower stiffness than that of the FNA, and because of which RTA could easily deform under the internal drying shrinkage stress, the specimens’ shrinkage length changes increased with the RTA content; however, the addition of PF decreased these length changes. Therefore, using 5%, 10%, 15%, and 20% RTA as a substitute for FNA increased shrinkage length at 28 days by 10%, 14%, 21%, and 27%, correspondingly.A total of 10% of RTA was the optimal amount to maximize bulk resistivity. With the addition of 2% PF and 10% RTA, this characteristic was improved by roughly 12%. It is worth noting that the ideal rate to achieve the highest possible bulk resistivity was established based on the amounts of PF and RTA employed in this investigation. Additionally, SF significantly improved bulk resistivity, particularly when more than 10% RTA was utilized;By adding RTA, the length growth under alkali-silica reaction assault was significantly reduced. Additionally, by adding more RTA, the impact of PF on the alkali-silica response development decreased, and there is no discernible alteration between the length growth of the sample with 2% PF reinforcement and concrete with 1% PF and 20% RTA.According to the materials utilized in this investigation, 10% is the ideal amount of RTA to maximize the durability factor before it begins to significantly decline. Additionally, the durability factor increased by 13%, 31%, and 37%, respectively, when 2% PF was added together with 0%, 5%, and 10% RTA compared to the control mix.Up to 50 freeze-thaw cycles resulted in a momentous change in the specimens’ length. Due to ongoing cement hydration, autogenous shrinkage happened when the specimens were kept in the freeze-thaw chamber. At low cycle counts, the length changes of the specimens abruptly decreased and then gradually increased. The specimens’ length reduction also got worse as PF content increased. Alternatively, it declined as the RTA content increased, and the length changes recovered less as the RTA increased.

## Figures and Tables

**Figure 1 materials-15-08043-f001:**
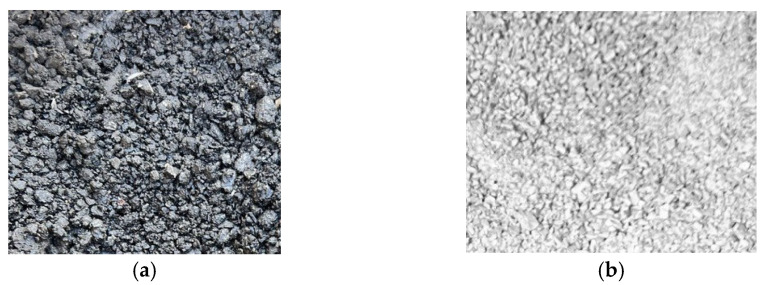
Utilized aggregates: (**a**) RTA and; (**b**) FNA.

**Figure 2 materials-15-08043-f002:**
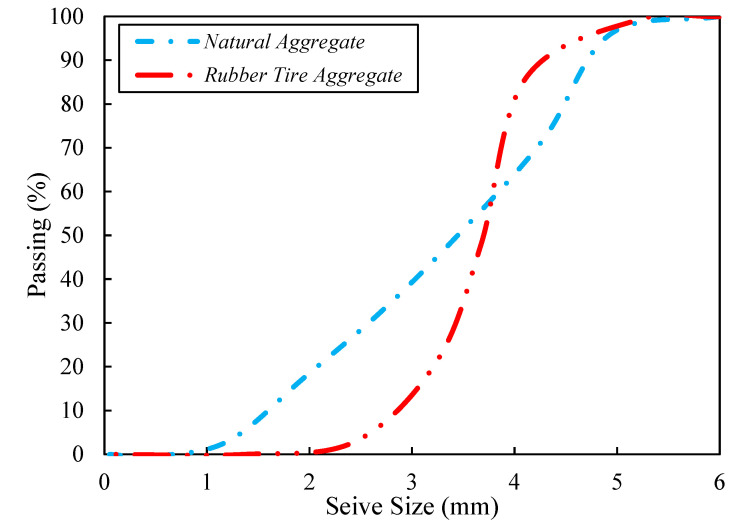
Used aggregates size distribution.

**Figure 3 materials-15-08043-f003:**
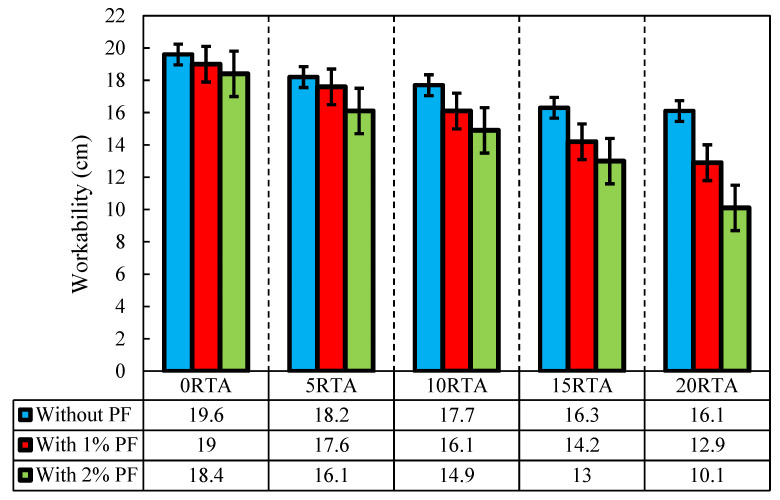
Effect of PF and RTA on the workability of samples.

**Figure 4 materials-15-08043-f004:**
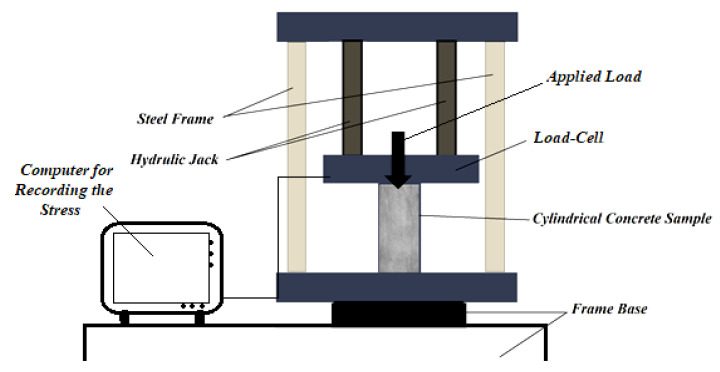
Used setup to determine compressive and tensile resistances.

**Figure 5 materials-15-08043-f005:**
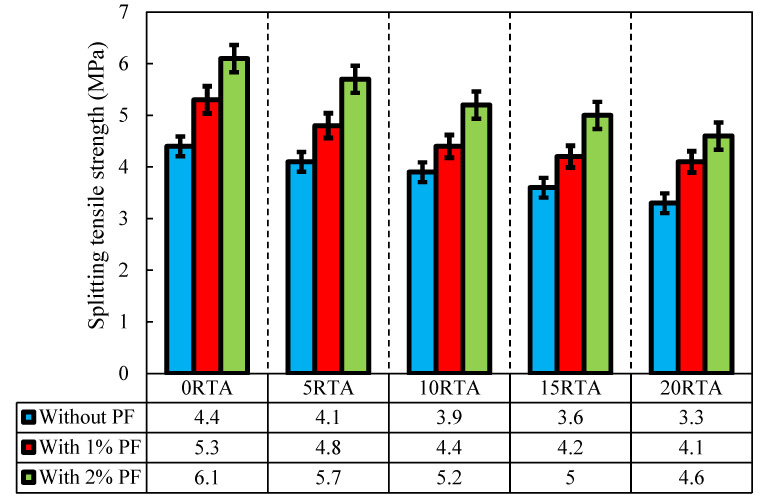
Impact of PF and RTA on the tensile resistance.

**Figure 6 materials-15-08043-f006:**
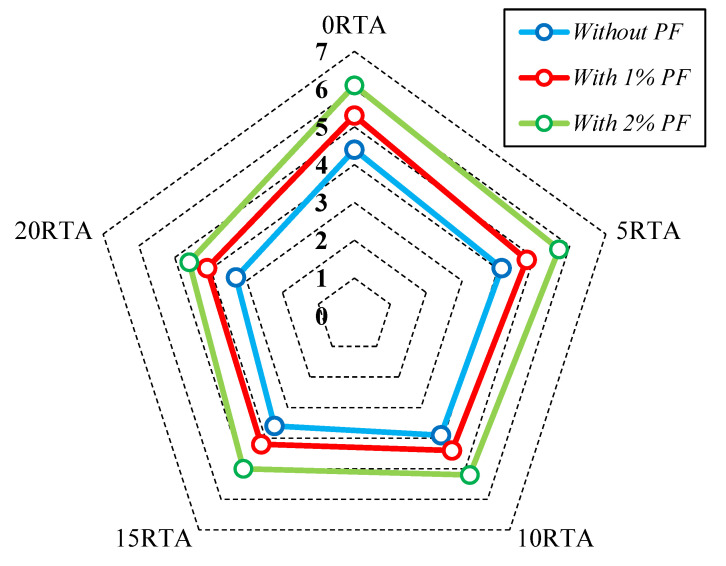
Influence of PF and RTA on the distribution of tensile resistance.

**Figure 7 materials-15-08043-f007:**
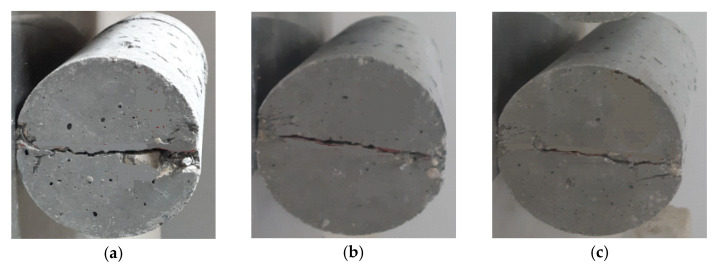
Failure in specimens due to tensile stress in the case of: (**a**) without RTA and PF; (**b**) without PF and with 20% RTA and; (**c**) with 2% PF and without RTA.

**Figure 8 materials-15-08043-f008:**
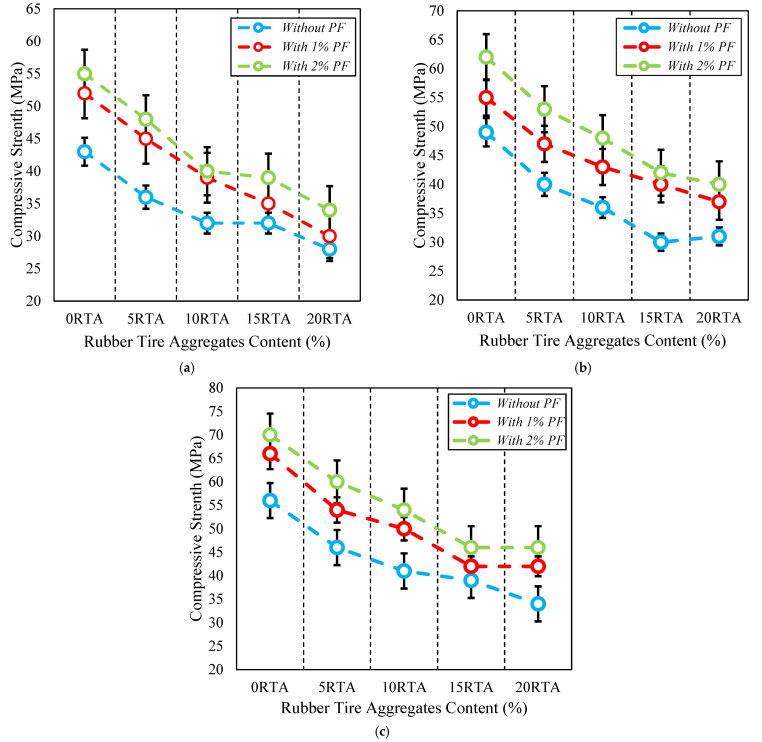
Compressive resistance of concrete having various PF and RTA fractions at: (**a**) 7 days; (**b**) 28 days and; (**c**) 56 days of curing.

**Figure 9 materials-15-08043-f009:**
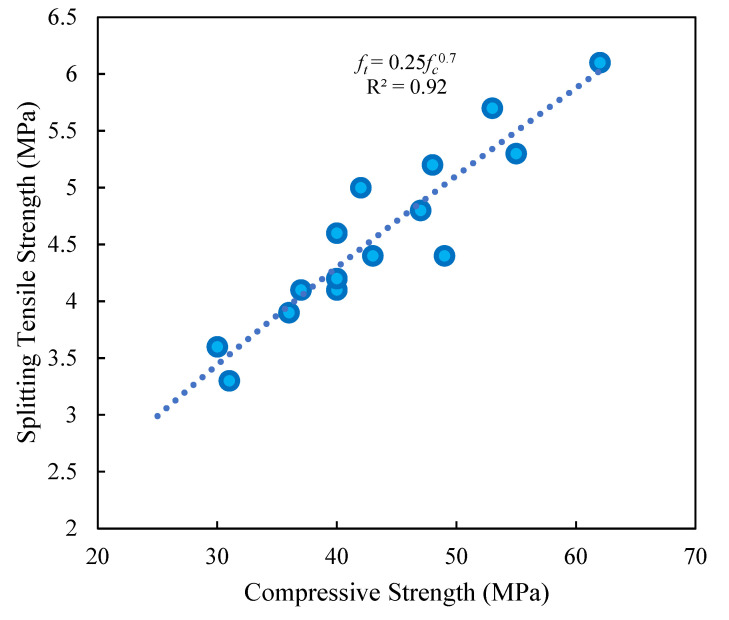
Relationship among compressive and tensile resistances of concrete containing different PF and RTA fractions.

**Figure 10 materials-15-08043-f010:**
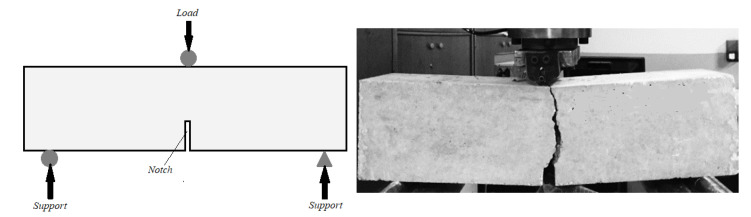
Fracture energy test arrangement.

**Figure 11 materials-15-08043-f011:**
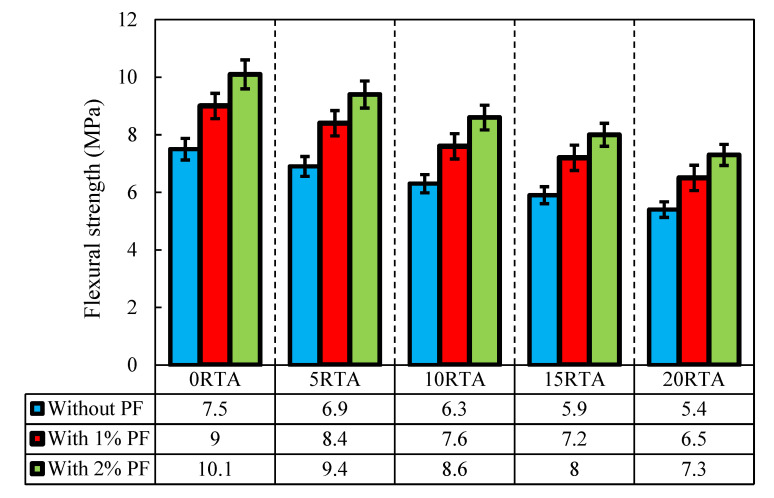
Impact of PF and RTA on the flexural resistance.

**Figure 12 materials-15-08043-f012:**
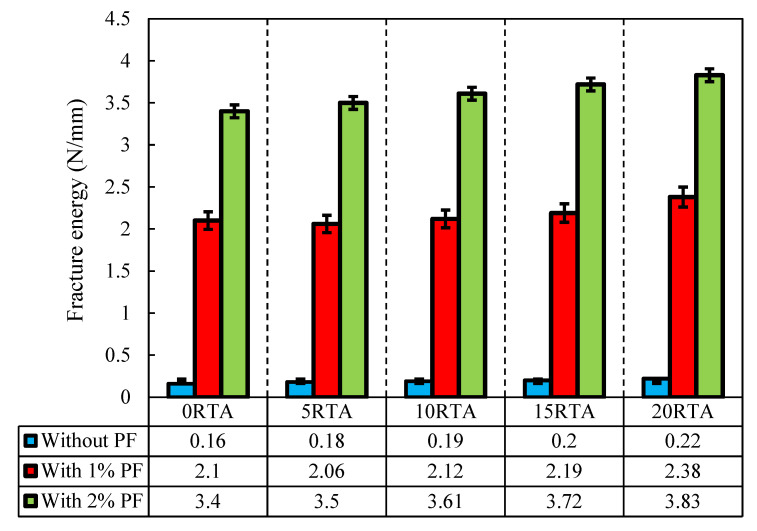
Effect of the PF and RTA on the fracture energy.

**Figure 13 materials-15-08043-f013:**
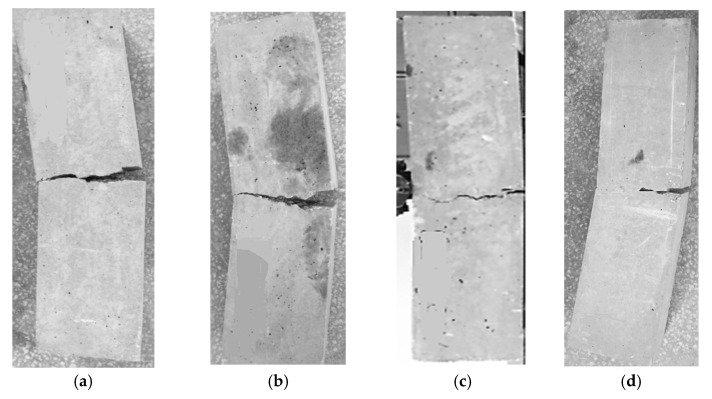
Failure in specimens due to flexure stress in case of: (**a**) without RTA and PF; (**b**) without PF and with 20% RTA; (**c**) with 2% PF and without RTA and; (**d**) with 2% PF and 20% RTA.

**Figure 14 materials-15-08043-f014:**
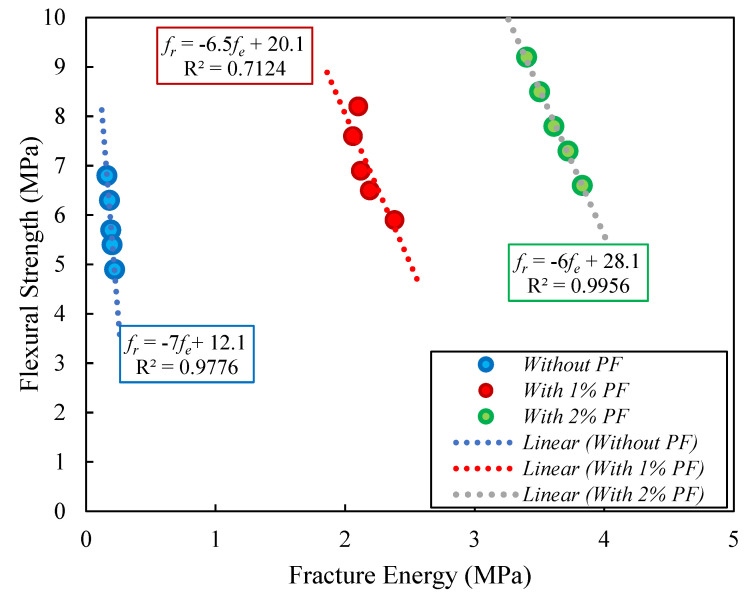
Relation among flexural resistance and fracture energy of sample containing different PF and RTA fractions

**Figure 15 materials-15-08043-f015:**
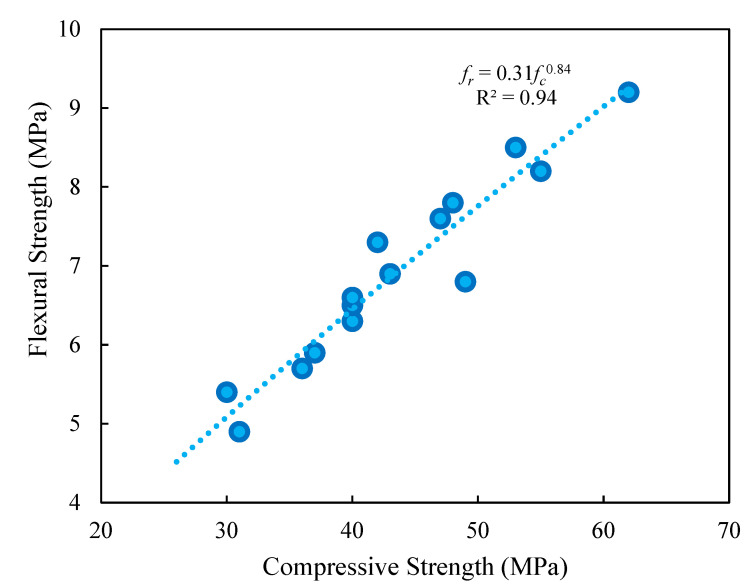
Relationship between compressive resistance and fracture energy of concrete containing different PF and RTA fractions.

**Figure 16 materials-15-08043-f016:**
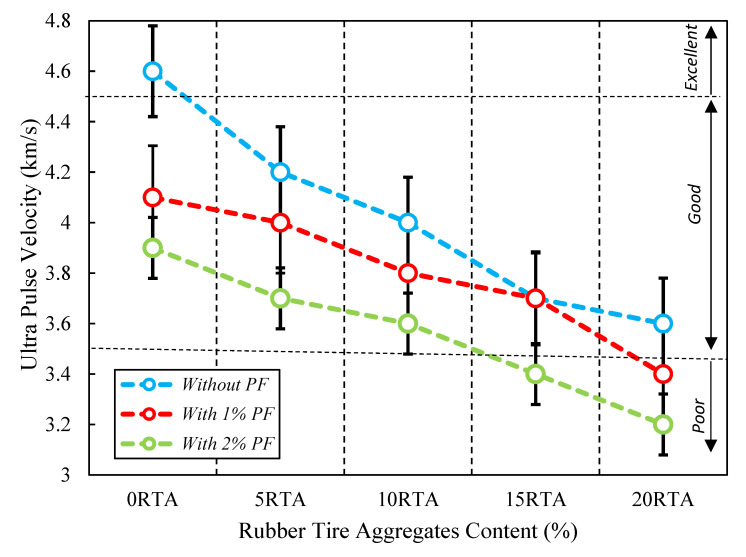
Effect of PF and RTA on the ultrasonic pulse velocity.

**Figure 17 materials-15-08043-f017:**
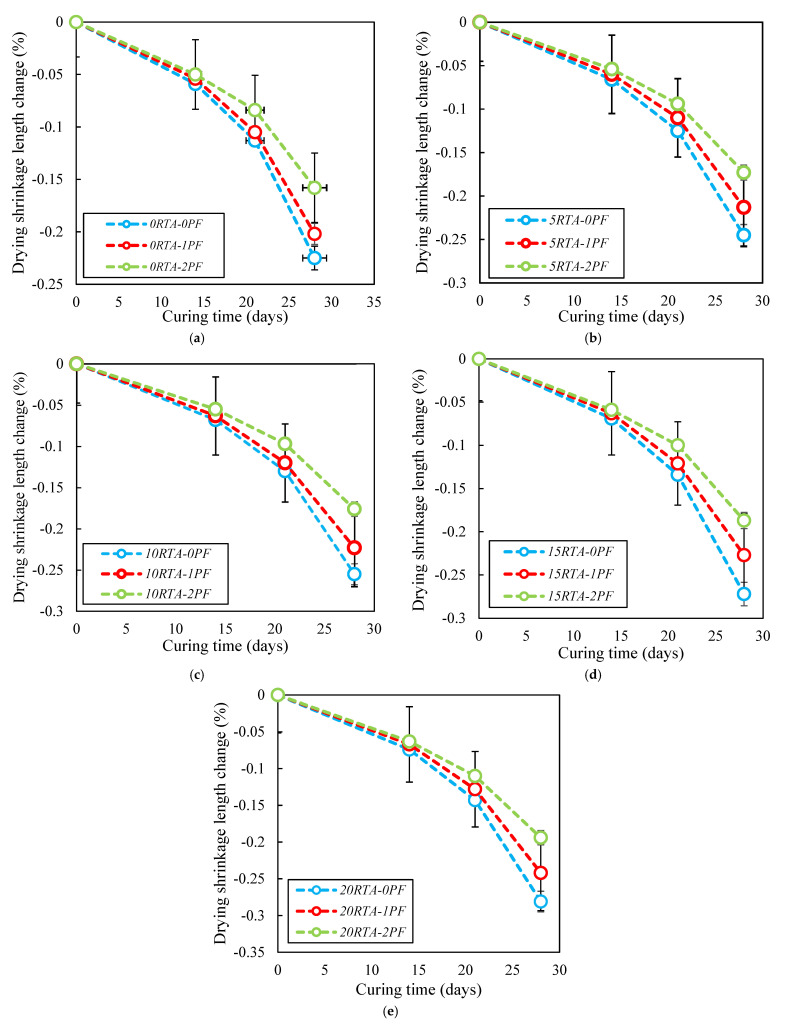
Influence of PF on the length variations due to shrinkage having: (**a**) 0% RTA; (**b**) 5% RTA; (**c**) 10% RTA; (**d**) 15% RTA and; (**e**) 20% RTA.

**Figure 18 materials-15-08043-f018:**
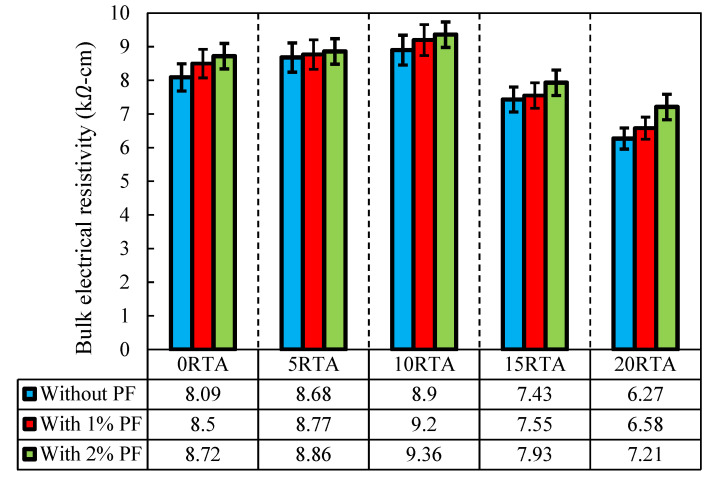
Impact of PF and RTA on the bulk electrical resistance.

**Figure 19 materials-15-08043-f019:**
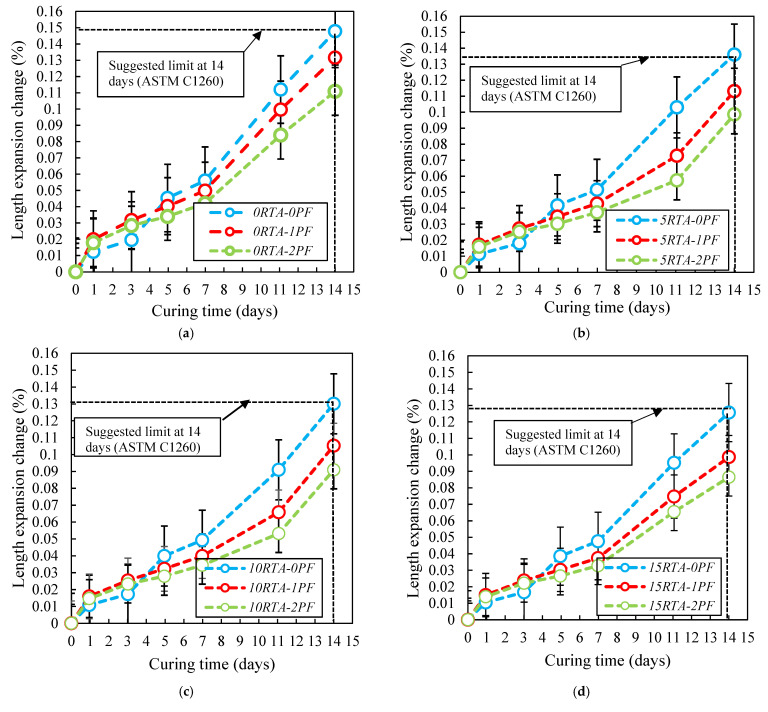
Influence of PF and RTA on the length growth. (**a**) without RTA, (**b**) with 5% RTA, (**c**) with 10% RTA, (**d**) with 15% RTA and (**e**) with 20% RTA

**Figure 20 materials-15-08043-f020:**
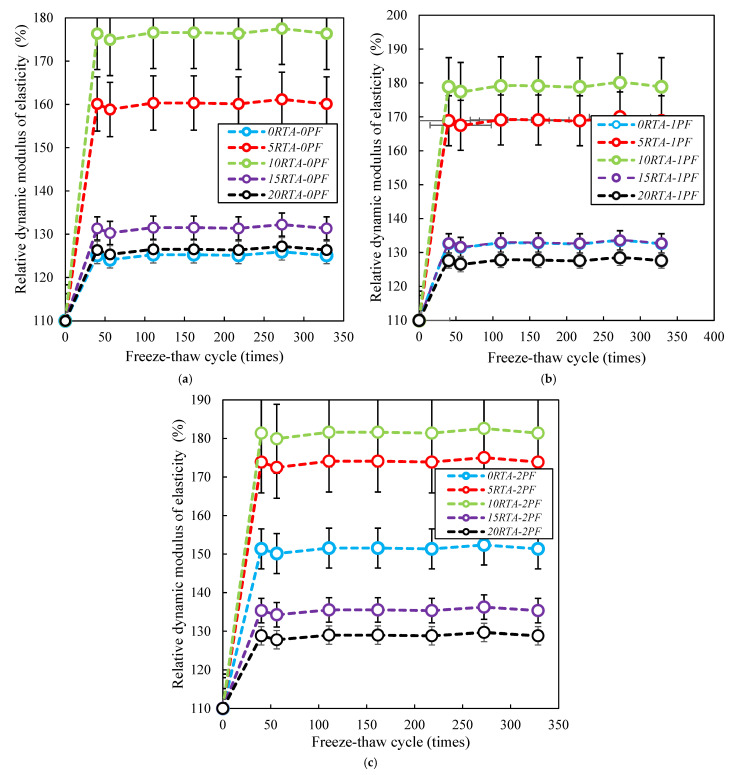
Dynamic elastic modulus of concrete having PF and RTA. (**a**) without PF, (**b**) with 1% PF and (**c**) with 2% PF

**Figure 21 materials-15-08043-f021:**
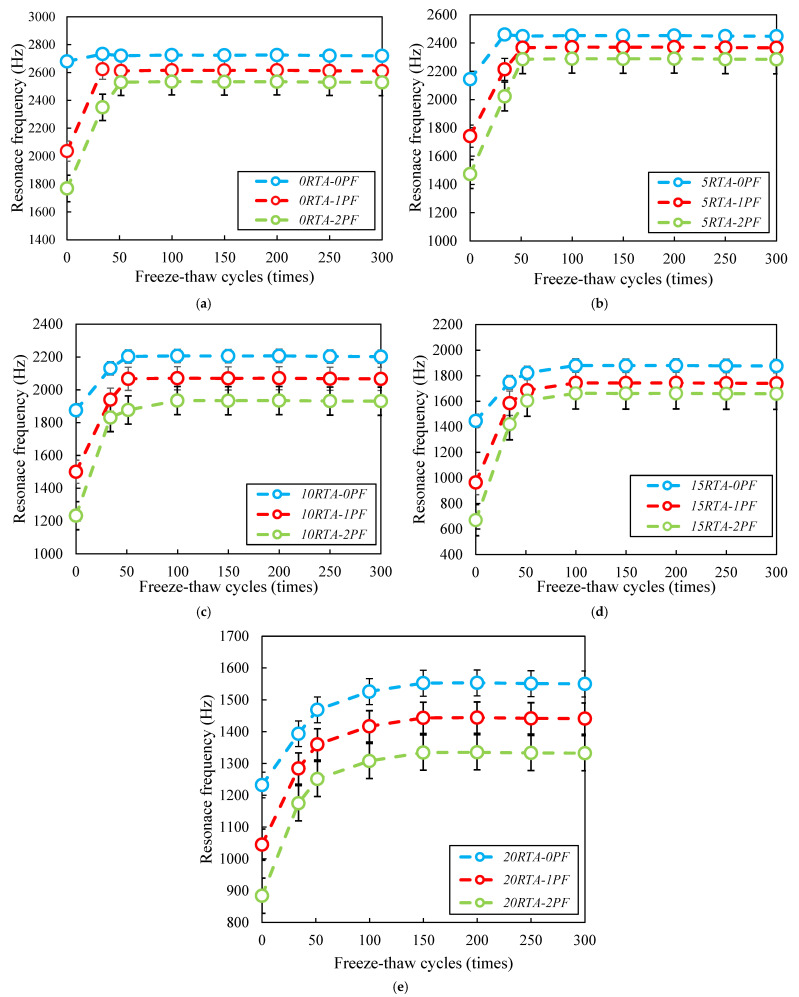
Influence of PF and RTA on the frequency: (**a**) Without RTA; (**b**) 5% RTA; (**c**) 10% RTA; (**d**) 15% RTA and; (**e**) 20% RTA.

**Figure 22 materials-15-08043-f022:**
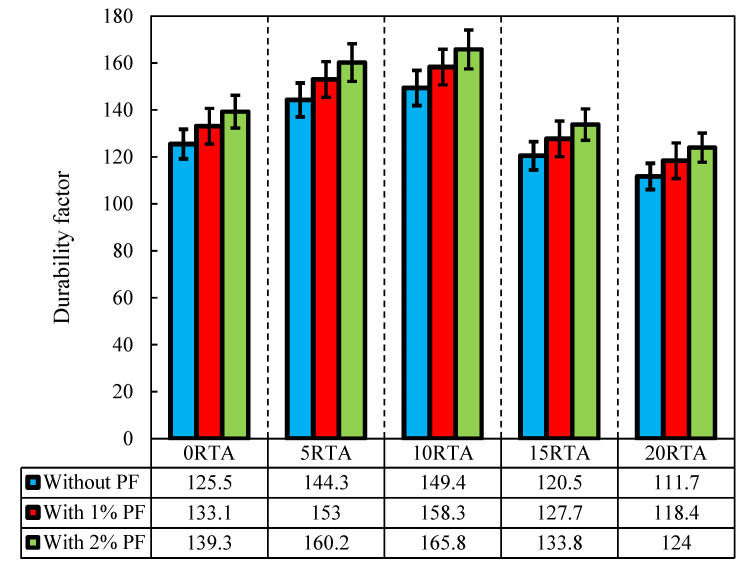
Durability factor of the concrete having different PF and RTA fractions.

**Figure 23 materials-15-08043-f023:**
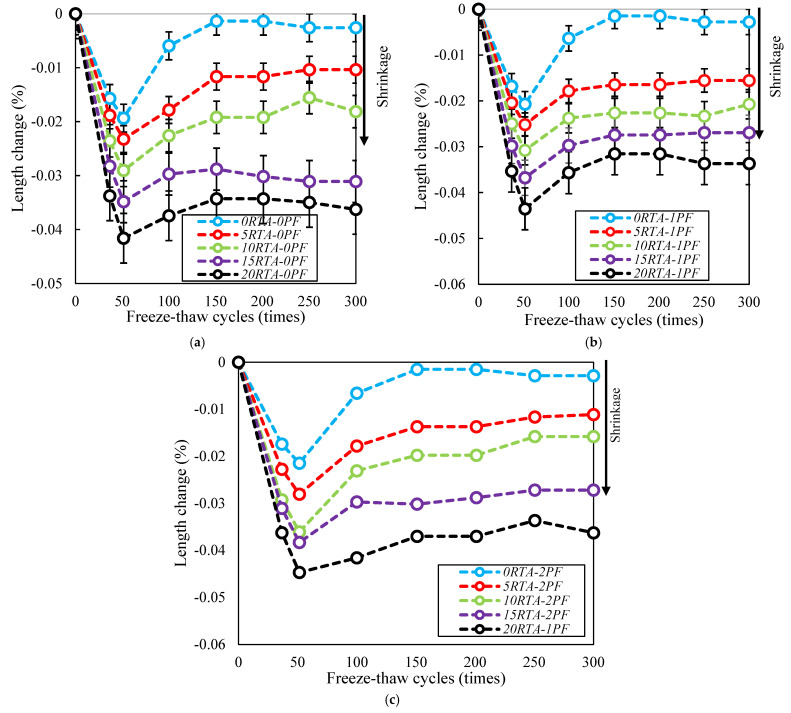
Change the length of the concrete mixtures having different PF and RTA fractions. (**a**) without PF, (**b**) with 1% PF and (**c**) 2% PF

**Table 1 materials-15-08043-t001:** Characteristics of the used cement.

Physical	
Feature	Value
Final set period	190 min
Initial set period	146 min
Exact surface	3215 cm^2^/g
Precise gravity	3.20 g/cm^3^
Autoclave extension	0.07%
**Chemical**	
$SiO_{2}$ (%)	22.34
$Al_{2}O_{3}$ (%)	5.12
$Fe_{2}O_{3}$ (%)	3.27
CaO (%)	58.29
MgO (%)	2.46
$SO_{3}$ (%)	1.97
$Na_{2}O$ (%)	0.36
$K_{2}O$ (%)	0.58
$C_{2}S$ (%)	1.34
$C_{3}A$ (%)	0.24
$C_{4}AF$ (%)	2.34
Free CaO	1.69

**Table 2 materials-15-08043-t002:** Characteristics of the used aggregates.

Fine Aggregates	
Feature	Value
Fineness moduli	2.65
Precise gravity	2.81 g/cm^3^
Water absorption	1.69%
Extreme size	4.74 mm
**Coarse Aggregates**	
Precise gravity	2.37 g/cm^3^
Water absorption	0.41%
Ultimate aggregate size	11.4 mm

**Table 3 materials-15-08043-t003:** Concrete mixes design (kg/m^3^).

Mixes	Cement	PF	CAN	FNA	RTA	Superplasticizer	Water/Cement
0RTA-0PF	450	0	950	815	0	1.5	0.47
0RTA-1PF	450	89	950	815	0	2.0	0.47
0RTA-2PF	450	178	950	815	0	2.0	0.47
5RTA-0PF	450	0	950	797	18	2.0	0.47
5RTA-1PF	450	89	950	797	18	2.1	0.47
5RTA-2PF	450	178	950	797	18	2.2	0.47
10RTA-0PF	450	0	950	779	36	2.2	0.47
10RTA-1PF	450	89	950	779	36	2.2	0.47
10RTA-2PF	450	178	950	779	36	2.2	0.47
15RTA-0PF	450	0	950	760	54	2.2	0.47
15RTA-1PF	450	89	950	760	54	2.3	0.47
15RTA-2PF	450	178	950	760	54	2.3	0.47
20RTA-0PF	450	0	950	742	72	2.3	0.47
20RTA-1PF	450	89	950	742	72	2.3	0.47
20RTA-2PF	450	178	950	742	72	2.3	0.47

## Data Availability

Some or all data, models, or codes that support the findings of this study are available from the corresponding author upon reasonable request.

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
