# Peer review of "Impact of Polypropylene Fibers on the Mechanical and Durability Characteristics of Rubber Tire Fine Aggregate Concrete"

_materials, 2022, doi:10.3390/ma15228043_

Round 1

Reviewer 1 Report

Dear Authors,

I have read manuscript titled: “Impact of Polypropylene Fibers on the Mechanical and Durability Characteristics of Rubber Tire Fine Aggregate Concrete” with great attention.

In my opinion, the article has a good scientific level and can be published after MINOR REVISIONS, because it is an original and valuable work, but I have some remarks to manuscript preparation.

Some minor issues to be considered by the Authors:

Line 184 - From Fig. 4 it's not clear how the load was applied, the rate and the measuring procedure. It is suggested to provide more detailed picture for the experimental set-up, if the picture is missing it can be replaced by the scheme.

Line 191 - Figure 4, shows only hydraulic jack, and can't be referred as test set-up, as it doesn't include any specimen and measuring equipment.

Line193 - What is the key diffidence between the information provided in Fig. 5 and 6?

Line 195 - Figure 7 shows the failure of the specimens rather than provides the information in regard to concrete reduction. The reference requires more detail explanation how the failure modes depicted in Fig.7 can be associated with declared concrete reduction.

Lines 207-208 - It should be explained how splitting tensile strength depends on the compressive strength relationship to the rubber tire aggregate content illustrated in Fig.8.

Lines 226-227 - The statement requires validation.

Lines 240-242 - As experimental investigation has been performed, it is suggested to support the statements with the figures of the corresponding failures.

Line 247 - Equation 2 requires explanations.

Line 300 - Figure 13 shows the failure of the specimens either with RTA or PF but not the specimens having RTA and PF, it is suggested to include the specimen with the combination of RTA and PF.

Lines 328-329 - Authors declare the decrease of ultrasonic pulse velocity of 21% to 31%, the statement of slightly reduced modulus of dynamic elasticity requires more detailed explanation and validation, especially when declaring that this is an originality of the study.

Line 356 - Figure 18 doesn't show any samples.

Line 395 - The Figure 20 shows not the values of modulus of dynamic elasticity, but the relative modulus of dynamic elasticity. Moreover, the term of relative modulus of dynamic elasticity has to be detailed.

Line 397 - Should be reference to Figure 20.

Line 399-400 - Incomprehensible statement (“When compared, …”), has to be rewritten.

Figure 20 - The Figure 20 (b) shows 4 curves, although the legend denotes 5. The curve 15RTA-1PF is missing. Overall the dashed curve notations are very confusing it is suggested to use more distinctive notations.

Reviewer 2 Report

The paper gives useful information on large number of parameters characterizing concrete containing rubber tire aggregates  and polypropylene fibers. Optimal compositions with respect to different parameters and as a whole are pointed out.

The paper has to be carefully read and attention to be paid to some unclear/unfinished sentences, repetitions, as well as to numbers of figures and description of the used/drawn formulas.

Detailed comments are available in the attached file.

The article would be improved and its overall merit increased if information is given on the chemical composition of the rubber used and especially - after its treatment with NaOH solution

Reviewer 3 Report

In this paper, the authors characterized the durability and mechanical properties of rubber tire fine aggregate concrete. Plenty of experiments were conducted. Overall, the paper provides impressive investigation of the properties of the RTA concrete. However, the writing style and presentation should be improved extensively.

 1.     Section 2 is lack of logic. It is difficult for the reviewer to distinguish the disadvantages of literatures and significance of presented work.

2.     The resolution of Figure is low.

3.     The style of the same type figure should be the same. For example, Figure 3, Figure 5, Figure 11 and et al. are bar charts while they look different on the expression of numbers. Figure 8, Figure 16, Figure 17 et al. are line chart while some of them are color figures, some of they have numbers.

4.     Cite the right figure. For examples, citation of figure in Line 195, Line 207 and Line 356 seem to be wrong.

5.     There are typo problems: Line 211 and Line 226 “TRA”, Line 300 “flaxure”.

 There are also some concerns on the contents:

1.     As displayed in Figure 2, there are large discrepancies on the gradation of RTA and FNA, will this affect the performance of samples?

2.     What does Figure 6 mean? How do the authors quantify the influence of PF and RTA on the distribution of tensile resistance, saying the value of the contour?

3.     Could the authors provide error bars in the bar charts and line charts to exclude the influence of errors that might be introduced in during sample preparation and testing?
